# American Foulbrood—Old and Always New Challenge

**DOI:** 10.3390/vetsci10030180

**Published:** 2023-02-23

**Authors:** Kazimir Matović, Aleksandar Žarković, Zoran Debeljak, Dejan Vidanović, Nikola Vasković, Bojana Tešović, Jelena Ćirić

**Affiliations:** 1Veterinary Specialized Institute Kraljevo, Zicka 34, 36000 Kraljevo, Serbia; 2Institute of Meat Hygiene and Technology, Kaćanskog 13, 11000 Belgrade, Serbia

**Keywords:** American foulbrood, etiology, diagnosis, control, *Apis mellifera*

## Abstract

**Simple Summary:**

The American foulbrood (AFB, foulbrood, American bee rot, severe bee rot, *Pestis apium*) fatal brood infection is still among the most deleterious bee diseases. Its etiological agent is the Gram-positive, endospore-forming bacterium *Paenibacillus larvae* (*P. larvae*). This review will place recent developments in the study of *P. larvae* into the general context of AFB research. We hope that this review will be helpful to beekeepers and indirectly to the apiculture industry.

**Abstract:**

American foulbrood (AFB) is exclusively an infectious disease of honey bee larvae (*Apis mellifera*) and their subspecies that is spread easily and rapidly and is often present in apiaries. Due to the resistance and pathogenicity of the bacterial causative agent of the disease, which has considerable epizootiological and economic significance for beekeeping, AFB was classified as a highly dangerous, infectious animal disease by the World Organization for Animal Health (WOAH). Considering the severity of the infection, a frequent occurrence, rapid and easy spread, epizooty and enzooty are common. We tried to present an overview of the latest information related to AFB through several chapters. In addition to the latest data on the etiology of the causative agent, the most important elements of the clinical signs of the disease are also listed. Along with an overview of classic microbiological and the latest molecular methods of diagnosis, we also discuss AFB treatment from its differential diagnostic aspect. We hope that through demonstrating the mentioned preventive measures and measures of good beekeeping practice, the review will contribute to the preservation of the health of bees and therefore the overall biodiversity of the planet.

## 1. Introduction 

American foulbrood (AFB; foulbrood, American bee rot, severe bee rot, *Pestis apium*) is exclusively an infectious disease of honey bee larvae, *Apis mellifera* and their subspecies, that is spread easily and rapidly, and it is often present in apiaries [1,2,3,4,5]. The infection of honey bee larvae initially takes place via the endospores during the first 36 h after hatching. Upon ingestion, the endospores begin to germinate and grow rapidly in the larval midgut for several days. Then, they destroy the midgut peritrophic membrane and epithelium. This opens a path for the bacteria to enter the hemocoel, resulting in larval death [5].

After detecting the disease, many measures and procedures must be implemented to stop it from spreading [2,6,7,8]. The disease is significant for international trade, so the export and import of bee colonies are allowed only under specific and controlled conditions [8,9]. WOAH classified AFB as a highly dangerous, infectious animal disease due to its significant epizootiological and economic significance for beekeeping. The causative agent of the disease was first isolated on the North American continent by an American scientist, White, in 1906 [10]. AFB is present in a considerable number of countries on all continents where honey bees and their subspecies are bred [2,8,11]. Considering the severity of the infection, its frequent occurrence, and rapid and easy spread, epizooty and enzooty are common [12]. As it spreads rapidly on the continental level, the disease has a panzootic character [10,12,13]. There is not much data on the prevalence of AFB agents at both regional and national levels, except those published by Morrissey et al., [9,14,15,16,17]. Since AFB causes considerable direct and indirect damage, radical measures are usually employed for prevention, suppression, and eradication. These are performed by notifying relevant authorities and stamping out any infected bee colony (i.e., destroying the bees and the hives) [2,4,6,7,8]. Taking into account the importance of beekeeping, this produces substantial economic and long-term biological losses since the intensive agricultural production of plants and direct and indirect production of food is impossible without bees as pollinators [2,11]. Additionally, beekeeping products occupy an important place in the nutrition, pharmaceutical, and cosmetic industries [18,19].

The economic value of the direct “operation” of honey bees, in the European Union (EU), is estimated at EUR 14.2 billion per year [20]. Smith et al. published data which indicate that due to the lack of pollinators, humanity loses between 3% and 5% of the production of fruits, vegetables and nuts [21]. This leads to 427,000 deaths per year, on a global level, as a result of lack of healthy food and the diseases caused by that.

## 2. Etiology of the Causative Agent of the Disease

The vegetative form of the causative agent of AFB is the *Gram*-positive, mostly mobile, rod-shaped bacterium, genus *Paenibacillus*, species *Paenibacillus larvae* (*P. larvae*)*,* formerly known as *Bacillus larvae* White [10]. In microscopic preparations of newly dead larvae, the bacilli are mostly individual, while in cultures, depending on the phase of growth on artificial nutrient media, they are present in the form of short or long chains. Endospores of *P. larvae* appear the most in older pathological processes, larvae degradation (characterized by an amorphous, glutinous, dilatable mass that dries to a scale) [11,22,23] and in older cultures on artificial culture medium (Figure 1). There can be over one billion endospores in one cell of a diseased brood comb and in a single dead larva [2]. The endospores are elliptical, mostly central, highly resistant to extreme temperatures and other physicochemical agents, and they are the only infectious cause of AFB [4]. In normal conditions and in pathological material, the remains of dried larvae, scales, old comb in the bee hive, and endospores remain infectious for up to thirty-five years, and endospores released from wax, propolis, honey, and other honey bee products remain so for over seventy years [24].

By applying molecular techniques, several genotypes of *P. larvae* can be confirmed [17,22]. The genotypes ERIC I and ERIC II of *P. larvae* have been the most commonly isolated genotypes in the world [2,17,25]. *P. larvae* genotypes ERIC III and ERIC IV have not been isolated for a long time, but there are several isolates in bacterial culture collections [2,22]. According to Beims et al. [26], a new ERIC genotype has been recently discovered—*P. larvae* ERIC V—in samples of Spanish honey (Table 1). All five genotypes have different forms of endospores. Morphologically, the endospores of genotypes ERIC I and II have a smooth surface, while the endospores of genotype ERIC III–V have longitudinal ridges [26]. All of those are pathogenic for bee larvae, give the same clinical picture (amorphous, glutinous, dilatable mass that dries into a scale), but are different because of the bacterial colony appearance, pigmentation, hemolysis, fermentation of carbohydrates, and virulence [2,4,22,25,26,27,28]. 

It has been shown that different genotypes of *P. larvae* have different virulence. While the ERIC I genotype results in a 100% death rate of infected larvae in 10–12 days, the ERIC II–IV genotypes kill infected larvae in 6–7 days, and the ERIC V genotype kills larvae in 3 days [2,3,23,26]. On suspecting AFB, veterinarians and beekeepers look inside the cells for a dilatable, glutinous mass as the main sign of disease [30]. In the case of larvae infected with *P. larvae* endospores of the ERIC II genotype, there can be lower numbers of or only several changed (larvae die earlier so that nurse bees clean them earlier) brood comb cells, with fewer characteristic signs, and hence, clinical examination can result in false negative findings for AFB [11,27,29]. Therefore, in the diagnosis of AFB, clinical signs must be combined with the selection of appropriate laboratory methods, all of this depending on the type, nature, and age of the material, that is, the type of examination (suspected disease or monitoring).

The primary sources of the disease are infectious, diseased, and dead larvae, as well as the scales of dried larvae containing a large number of endospores of *P*. *larvae.* Young bees, by cleaning, contaminate the food for feeding larvae, royal jelly, honey, and pollen in the beehive, comb, frames, and inner walls of the beehive. Since there can be over one billion endospores in one contaminated cell of the comb, every frame of comb with its diseased, infectious, and deceased bee brood contributes to the largest cause of the spread of infection [11,31]. The medium lethal dose (LD_50_ = endospore dose that can kill 50% of larvae) required for the commencement of infection is 10 or fewer endospores in a bee larva that is 12–36 h old, although it is often variable [2]. In natural conditions, the infection develops by feeding the larvae with food contaminated with endospores of *P. larvae*, a process performed by nurse bees [4]. Although all worker bee larvae, queens and drones in the bee brood are equally susceptible to infection, infection of queen and drone larvae is rare under natural conditions [32]. Bees that engage in hygienic behavior, i.e., removing diseased and dead larvae and preparing the queen to lay new eggs, play a leading role in the spread of disease within an apiary. The removal of pathogens is a type of immune response in bees (social immunity). Genotype *P. larvae* ERIC II kills infected larvae for 6–7 days (LD_100_ = 6–7), demonstrating high pathogenicity at the larval level. It indisputably confirms that larvae infected with genotype ERIC II will die before the capping of bee comb cells. This gives the bees enough time to remove the dead larvae and clean the cells. As a consequence, a smaller amount of *P. larvae* endospores is produced at the level of the bee colony. Genotype *P. larvae* ERIC I kills infected larvae in 12 days (LD_100_ = 12). Therefore, genotype ERIC I is less virulent at the individual larval level, but infected larvae certainly die after cell capping. Hygienic nurse bees find it difficult to remove dead larvae in such cases. As a consequence of the above, there is a possibility of the high production and spread of *P. larvae* endospores within the bee colony. As a consequence of the above, infection with genotype ERIC I causes rapid death of the bee colony, compared to genotype ERIC II, which causes delayed death of the infected bee colony [2,11,22,28,33]. The young worker bees play a role in spreading the disease inside the beehive [33]. By removing dead larvae from the of the bee brood and cleaning it, nurse bees spread the endospores throughout the hive. Through their regular and normal activities in society, bees contaminate food. The result is that the disease spreads through contaminated food to newly hatched, uninfected larvae. As the larvae age, their resistance to infection builds up. In about 12 h, endospores germinate in the intestines of the larvae. In the colonizing midgut, vegetative forms of bacteria reproduce without visible tissue damage and integrity of the epithelium of the midgut damage as long as the bacteria are in the “commensal” phase [2,23,28]. *P. larvae* are known to produce numerous virulence factors (SplA, *Pl*CBP49, Plx1, Plx2, C3larvin, and others) that are responsible for damage to the epithelial barrier of the intestine so that *P. larvae* pass into the hemocoel [34,35,36,37,38,39]. These virulence factors lead to adhesions *P. larvae* on midgut cells, chitin degradation, ADP-ribosylating of DNA and cell death without signs of inflammation and apoptosis [34,35,38].

The dried remains of the larvae (scales) are difficult for the workers to remove and they are a constant source of infection for the new brood. Infection can thus spread between colonies by swarming, which spreads the causative agent of AFB by drifting, robbing, and especially by beekeeping operations whereby contaminated material is transferred to healthy colonies [2,40,41]. As it is a bee brood disease, bees can also transmit/spread the causative agent of the disease by robbing honey and endospores of *P. larvae* found in bee excrement [16,40]. The disease is spread by bee migration to forage, using unsterilized beekeeping tools, watering bees in puddles and pools that can be contaminated with bee feces and *P. larvae* endospores (due to a lack of water devices in apiaries), or buying and selling bee colonies [2]. Insects (e.g., flies, ants, and wax moths) also play a substantial role in the spread of the disease [42]. The replacement of comb material which contains *P. larvae* endospores from dead larvae (i.e., inadequate beeswax processing), infrequent comb replacement, and supplemental feeding with honey that contains *P. larvae* endospores can lead to the spread of the disease [2,13]. 

Although adult bees do not contract the disease, bee colonies suffer from disturbed replacement of bee generations [2,11]. Older bees die and there are no new bee generations. As a consequence, the bee colony is constantly weakened and dies [2,4,11]. In these colonies, honey robbery and wax moths commonly lead to further spread of infection [11,13]. 

Early detection and AFB diagnosis, i.e., applying adequate and prescribed legal measures and procedures, are the only significant means to suppress and eradicate the disease [4,7,8,43,44]. 

## 3. Clinical Signs of the Disease

Clinical AFB signs are variable and they depend on the genotype that infected the larvae, the strength of the bee colony, and the eventual bee colony resistance to AFB [2,3,4,33]. However, clinical signs are most often not observed in an unsealed litter (before metamorphosis) because hygienic nurse bees recognize and throw out diseased and dead larvae during cleaning the cells of the bee brood (infection with endospores of *P. larvae* genotypes ERIC II-V) [22,23].

For a short time after capping the brood cells, changes occur in the color, structure, and consistency of the larvae. When we examine the colony, scattered clogged cells are noticeable, which are often located between the cells of the healthy brood. Since AFB is a disease of bee broods/larvae, the first clinical signs of the disease, although visible later than the changes in the larvae, are noticeable on cell caps. Changes in color, layout, and integrity of the caps usually occur twenty days after the larvae are infected. In larvae that die after closing, pathological processes develop later, and so the typical hygienic behavior of bees appears after changes in the caps. Microclimatic conditions inside the hive also affect diseased and dead larvae. The caps acquire a lemon color and then turn dark brown; they are wet, soft, and slightly concave [2,12,22,29,30]. Dark spots on their surface leave the impression of dirty, greasy brood combs. In addition to that, tiny holes with irregular edges, often scattered on the brim, are the result of the removal of dead larvae and cleaning bee brood combs. In autumn, when the queen bee stops laying eggs, the changes on the caps are the most prominent since only the infected brood remains. When cell caps are opened, it is noticeable that the color of the larvae changes from pearly white and distinctively glowing to a greyish-yellow that subsequently turns into a sandy color, then light brown, and finally dark brown (the color of chocolate). At the same time, larvae lose their specific form, and their bodies turn into a semi-liquid, glutinous, ropy mass [4]. At the beginning of the rotting process, resilience is high; therefore, the mass can be stretched in threads from a cell with a toothpick or a match for a few centimeters. The whole rotting process, post-infection, lasts from five to eight weeks. In the latter stages of the disease, due to cleaning (communal bee hygienic behaviour), evaporation and drying, the mass becomes dense, fitting closely to the cell wall; it looks like a black and brown scale, is pin-head sized, and is difficult to spot (in a process older than two months). In rare cases, the remains of dead larvae can be of aqueous non-specific content [30]. Figure 2 depicts the clinical picture and the development of pathological processes over time for AFB.

Larvae die depending on the genotype of *P. larvae* in a period of 3–12 days after infection (sometimes before the sealing of cells of the bee brood) [2,11,22,23,26,28]. Since the ERIC I genotype is less virulent than the other ERIC II-V genotypes, the majority of larvae infected with the ERIC I genotype die later. This occurs sometime after the bee brood cells are sealed, that is, after the onset of metamorphosis [2,25,28]. Nurse bees will open the cap, clean out the dead larvae, and leave an empty cell. Genotype ERIC II, as well as the remaining three genotypes, are more virulent and the largest number of larvae are killed before the onset of metamorphosis, i.e., before cell capping. These killed larvae are recognized earlier by the hygienic nurse bees and removed from the brood. The bacteria, *P. larvae*, kills the infected larvae. The sooner the hygienic nurse bees expel these larvae, the fewer vegetative forms of the bacteria will sporulate, i.e., spread more slowly within the colony. The degree of virulence of *P. larvae* ERIC I at the larval level is lower compared to the degree of virulence of *P. larvae* ERIC II, while at the colony level there is a reverse correlation [13,28]. In the presence of less virulent genotypes of *P. larvae*, (ERIC I), larvae and pupae die after bee brood bee comb cells are sealed [2,25]. If death occurs during the pupal stage, the projecting tongues of pupae can be one of the most characteristic disease signs, although very rare [30]. On the part of Figure 3A shows the projecting tongues of dead pupae as one of the few signs of AFB [30].

## 4. Control of the Disease

Somerville et al. [45] recommend *good agricultural practice* (GAP) and *good hygienic practice* (GHP) in beekeeping. These are prevention measures that include all general and preventive procedures in order to prevent the disease from getting into non-infected apiaries. In case of clinical signs or suspicion of AFB, it is the duty and legal obligation of everyone in the chain of biological and food production, from beekeepers to veterinarians in the field, veterinarians in the laboratory, and veterinary inspectors, to take appropriate measures and procedures. Good practices include: apiary hygiene; regular checks of the colonies; comb replacement; current disinfection while working in apiaries; inspection of purchased bee colonies and foraging colonies; using sterilized comb foundations produced according to *hazard analysis and critical control points* (HACCP) systems; laboratory inspection of supplemental feed; and special attention when accepting feral swarms [7,13,43,44,46]. 

The first and most important step in the suppression of AFB is early disease detection by beekeepers [15,46,47]. A thorough inspection is necessary in the spring if the disease has not been previously diagnosed. When symptoms of the disease appear, during the inspection of each colony, all beekeeping tools and equipment (beekeeping knife, gloves, blouse, etc.) that are used must be disinfected or during the inspection researchers must use disposable equipment that can be safely destroyed after the inspection (gloves). If the disease has occurred nearby, examination should be conducted more often and with the utmost caution. If the disease is detected in one colony in an apiary, all the colonies in that apiary must be examined, as well as those in apiaries that are located within a 3 km radius [6]. If AFB is suspected, the veterinary inspector must be informed immediately in order that all actions and measures needed to suppress and eradicate the disease are undertaken in a fast and efficient way [6,7,11].

AFB cannot be cured and so in accordance with legal procedures, in case of disease and by order of persons authorized by law, radical measures must often be taken. Namely, operators must close and inspect the infected apiary; close the hive entrance and suffocate the bees in the infected bee colony when all the bees have entered the hive (usually in the evening); burn the frames with a comb, as well as the honey and bees from the infected colony; bury all the mentioned materials and dead bees, and conduct thorough disinfection [8,10,43] or, less often, deploy the shaking bee hives measure [7]. After thorough mechanical cleaning, usable (but still new, preserved) hives are burnt with a blowlamp and disinfected with a 2–6% NaOH solution (caustic soda), KOH, Na_2_CO_3,_ or alkaline hypochlorite solution (5% NaClO in the above solutions), heated to 80 °C. After applying one of the aforementioned solutions, the hives are immersed for 5–15 min (this increases exposure to the equivalent of two hours of spraying), then washed with warm water and dried [43,44]. A variety of different studies [43,44] after mechanical cleaning suggested the need to disinfect tools and other equipment with a Na_2_CO_3_ solution (1 part Na_2_CO_3_ and 5 parts water) or a 0.5% hypochlorite solution (NaClO) for twenty minutes. Additionally, they suggest to disinfect the apiary area with a solution of 10–20% Ca(OH)_2_. Over the next two months, the remaining bee colonies must be monitored closely while enhancing the disinfection of all the tools used to inspect every hive. If there are no new clinically suspicious AFB colonies during this time or if the suspicious colonies are not confirmed as diseased after laboratory examination, there is considered is considered to be no infection [6]. One method still in use for the eradication of AFB in some countries is the shaking method [7]. Considering that AFB is a disease of bee brood, not adult bees, it means shaking/transferring adult bees from the infected community to new disinfected hives and bee combs free from *P. larvae* endospores in the rest of the beehives in quarantine. Everything that is infected in hives (wax, honey, pollen, propolis, litter) is destroyed. Special/other equipment is always used in the implementation of these measures [7].

AFB is first suspected based on clinical signs of the disease in the apiary, but the diagnosis of AFB is based on laboratory identification of the pathogen [2]. During preventive examinations (monitoring) for the presence of AFB causative agents in the laboratory, in the case of a positive diagnosis, the colonies and apiaries must be clinically examined and legal measures taken [4,7,8,44,47]. While dealing with material suspected of harboring AFB, all biosecurity measures, risk analyses and assessments must be applied to reach the standard for biorisk management in veterinary laboratories and animal facilities [8]. 

When collecting samples from a clinically suspicious and/or possibly diseased (infected) bee colony, it is sufficient to send a sample of the bee comb with suspicious cells to the appropriate laboratory for testing. If possible, the complete comb frame should be submitted for inspection in order to avoid damage during packaging and transport. Alternatively, part of the brood comb of at least 20 cm^2^, with visible changes and containing diseased and/or dead larvae can be submitted for inspection. An experienced clinician can collect the remains of dead larvae/pupae from the cell walls with a sterile swab, and thus, significantly facilitate the packaging and transport of materials to the laboratory. When AFB is confirmed by the laboratory in a bee colony, any bee colony located near an infected hive must be suspected as having the disease, and samples (honey, pollen, royal jelly, wax, food, adult bees) should be taken and sent to the appropriate laboratory for diagnosis. In order to prevent the spread of the disease from the infected apiary, samples of honey, comb, and bees can be used to detect AFB in bee colonies that show no clinical signs of the disease. Nowadays, routine sampling of bees and honey is increasingly used in regional control programs (monitoring) to detect endospores of the causative agent of AFB worldwide [8,14,15,16,30,45,47,48,49]. 

Samples of bee brood, bee carcasses, waste from the hive floor, and wax should be packed in paper, paper bags, newspaper, cardboard, and/or wooden box and delivered to an authorized laboratory for further examination. Packaging and shipping of this material in plastic bags, aluminum foil, wax paper, tin, and glass containers should be avoided, as these materials can damage the sample and make it inadequate for examination [8].

Although the vegetative forms of *P. larvae* are susceptible, due to the nature of the causative agent, the biology of bee development, the high resistance of endospores and the highly contagious nature of the disease, the use of antibiotics for both preventive and therapeutic purposes is prohibited. Their use will lead to the spread of diseases and beekeeping products will contain antibiotic residues. In spite of everything, in some countries such as the United States of America (USA) the use of antibiotics is allowed to control AFB and other diseases of bee colonies. Every organism, including the bee organism, has a normal, physiological bacterial microflora/microbiome (oral cavity, stomach, intestines) on it, around it, and in it, which protects it from pathogenic microorganisms. Uncontrolled, “preventive”, misuse of antibiotics will lead to the suppression of normal physiological microflora and the uncontrolled growth of fungi. Residues of antibiotics in honey and other bee products, during consumption, will have an adverse effect on the microbiome of humans. The most commonly used antibiotics in beekeeping are tetracycline, streptomycin, nystatin, and fumagillin. The use of antibiotics is prohibited in beekeeping in the EU and the Republic of Serbia [2,4,39,45,50,51]. For epizootiological reasons, honey from the hives in diseased colonies should not be used. During honey extraction and consumption, the disease will spread, and endospores can pass through the body unaltered [8,12,46].

## 5. Differential Diagnostics of the Disease

At first glance, clinical symptoms AFB can be mistaken for sacbrood disease, European foulbrood, and *varroosis*, mostly in the early spring [8,12,30,46,52]. Sacbrood disease is a viral disease of the bee brood, mostly benign, in which larvae do not pupate. Unlike AFB (in which dead larvae transformed into an amorphous, gelatinous and extensible mass and, at the end of the process, often in the spring, into a scale), in the case of sacbrood disease, dead grey-brown larvae under the cap are easily pulled out of the cell and take the shape of a bag (sac), Figure 3B. If the sacbrood process is older, due to drying, the larva turns into a scale, and the head and the last part of the body bend, and thus, it forms into a boat shape. In the early spring, with *varroosis*, under the cap of the cells of the bee brood, one can see one of the deaths of the often deformed developmental forms of the bee (pupa, young bee) and the presence of *Varroa destructor* that is most often united with viruses [12,46].

European foulbrood is both an uncapped and sealed brood disease. As such, so diseased larvae die in both uncapped and sealed broods. The causative agent of the disease is a bacterium of the genus *Melissococcus*, species *Melissococcus plutonius* (*M. plutonius*). Diseased larvae are yellowish, often bloated, limp, mushy or liquid; when pulled out of the cell, their chitin layer tears. Sometimes, larvae stop being white due to drying, they lose their pearly glow and segmentation, they do not transform into pupae, and bees can easily eject them from the hive [8,12,30,46].

In early spring, dead bee colonies can often be found with a sealed, unhatched brood, and plenty of food in the hive. Sometimes a doubt can be raised about the cause of the death of the colony, i.e., AFB and/or *varroosis* [12,46]. In the case of *varroosis*, after opening the caps, there is usually a formed pupa or young bee in the cell with one or more varoa mites. When it comes to AFB, there are no completed stages of bee development (pupae, young bee), but it is a late stage of the disease (the mass is dried out), and the cell is apparently empty. However, if we examine it carefully, we can find black and brown scales at the bottom, which is quite common for AFB disease processes older than two months [46]. WOAH has presented a broad outline of various diagnostic methods. However, due to differences in sensitivity, the most appropriate of the described methods should be selected [8]. In addition to classical microbiological and antibody-based techniques, molecular and physicochemical methods are used. A wide range of samples can be delivered to the laboratory (diseased and dead larvae, cell swabs, honey, pollen, royal jelly, wax, dead bees, food, and debris from the hive floor) [8,9,46]. The specific methodology applied depends on the type and nature of the materials delivered to the laboratory and the purpose of the examination (preventive examination, suspected disease, already confirmed presence of the disease) [9,16,25,46]. 

Within classical microbiological methods, microscopic determination of endospores of *P. larvae* and microbiological cultivation and isolation on nutrient media, with biochemical identification of the causative agent of AFB, are the most commonly applied methods [8,22,30]. There are several enriched selective media for *P. larvae* culture: *P. larvae* agar (PLA), Mueller–Hinton, yeast extract, glucose, KH_2_PO_4_, sodium pyruvate agar (MYPGP), brain–heart infusion medium supplemented with thiamine agar (BHIT), yeast extract agar (J-agar), and Columbia agar from sheep’s blood agar (CSA) [17]. Within the antibody-based techniques, the immunodiffusion test, fluorescent antibody technique (TFA), and enzyme-linked immunosorbent assay (ELISA) are applied [53,54,55,56].

In the identification of the causative agent of AFB, molecular techniques are applied. Polymerase chain reaction (PCR), real-time polymerase chain reaction (real-time PCR), and pulsed-field gel electrophoresis (PFGE) are the most commonly used in laboratories [2,4,16,22,25,30,46,47,49,57,58]. Real-time PCR analysis of the 16S rDNA gene of *P. larvae* represents an alternative, rapid diagnostic tool. As a part of scientific research work as well as genetic and epidemiological studies, methods of partial genome sequencing are more often applied in laboratory diagnostics. These are Multilocus sequence typing (MLST), multiple-locus variable-number tandem repeat analysis (MLVA), and high-throughput sequencing (HTS) [57]. It is also possible to detect *P. larvae* using microbiome analysis [59]. A method that can also be used to identify this pathogen is based on a physicochemical technique, namely the matrix-assisted laser desorption ionization time-of-flight mass spectrometry (MALDI-TOF MS) [60].

The microscopic method of detecting *P. larvae* endospores is definitely less sensitive than microbiological isolation, but both methods are less sensitive than PCR methods [2,47,48,61,62]. The presence of *P. larvae* can be determined by microbiological isolation and PCR methods from bee colonies in which the disease has not occurred [16,25,46,47,49].

The existence of the endolysin cell-binding domain (CBD), which binds *P. larvae* [63], may lead to new methods for the identification of bacterial strains that cause AFB. Yones et al. [64] investigated the possibility of applying of hyperspectral technology as a new trend for immediate detection of AFB disease in honey bee larvae. Early detection of AFB disease in honey bee colonies is certainly of substantial importance as interchanging of colony components can spread easily AFB to healthy colonies.

## 6. Prevention of the Disease

Legally prescribed measures must be implemented for the detection, monitoring, suppression, and eradication of AFB. Those are prescribed at national, regional and international levels [6,65]. Among other things, these include reporting diseases to competent authorities, conducting epidemiological studies, and monitoring the prevalence of diseases [6,7,8,11,65]. 

It is also a duty of all responsible people in the beekeeping production to regularly monitoring the health condition of bee colonies, to conduct inspections, to launch anti-AFB initiatives, to implement necessary legal measures, and to share any suspicions of AFB with the competent veterinary services [6,8,15,44,49,65]. When working in the apiary and with bee colonies, special attention should be paid to bee and colony health. It is necessary to conduct a detailed, comprehensive, and expert inspection of bees and bee colonies for diseases, with special attention paid to the presence of bee brood diseases, including AFB. This should be performed at least twice a year, in autumn, including sampling and culturing the bacteria (prior to wintering) and in spring (before bees begin to forage) [7,16]. The health status of apiaries or bee colonies should be observed before commencing any work. Every observed change in the brood, comb, or apiary, the beekeeper must take it seriously and, in case of occurrence, report it to the competent veterinary service. In that case, fast and reliable laboratory tests/diagnostics will be provided, which is a prerequisite for taking legal measures to prevent the spread, control, and eradication of the disease. Some suspected diseases of bees and bee broods, especially AFB, can only be confirmed by laboratory analysis. Therefore, it is obligatory to regularly perform diagnostic testing of the materials originating from any apiary that suspects AFB [30,64,66,67].

In the case of the AFB outbreak, all provisions that refer to measures and procedures in the early detection, monitoring, prevention, suppression, and eradication of the disease must be comply [4,7,8,44,65,68]. Bee production should be organized into associations, because this is the best way fight against the negligence of individual beekeepers. All participants in the process of bee production (beekeepers, veterinarians, fruit growers, agronomists, ecologists) must be educated through meetings, sharing experiences, and expert lectures. Individual beekeepers, beekeeping organizations, clinical veterinary services, laboratories, and veterinary inspectors must all work together to achieve this goal [68].

Figure 4 shows the destruction of infected bee colonies by burning and burying them to render them in order to implement a part of the measures aimed at eradicating the diseases. The disinfection of apiaries has also been shown to be another measure in the chain of disease eradication.

In some segments of organized bee production (e.g., wax and comb production, packaging and marketing honey to third parties, bee feed production), besides using GAP and GHP measures, HACCP systems must also be in place [68,69,70]. A fundamental principle behind implementing these measures is to enhance the quality of food and feed that will be produced in accordance with physico-chemical quality parameters and be microbiologically safe [70,71]. 

In addition to the above, one of the measures to control AFB is to determine the presence of *P. larvae* endospores in honey and wax samples (honey wax comb foundation) and, in particular, to implement monitoring programs for honey and honey bee samples to detect *P. larvae* endospores [9,19,25,43,49,61]. This latter approach is certainly efficacious when the greater costs of suppressing and eradicating AFB compared to the costs of implementing a national AFB control program are taken into account. Additionally, the eradication of AFB can be extremely difficult since the endospores of *P. larvae* are particularly resistant to the prescribed chemico-physical disinfection measures. Thus, any remaining endospores are viable and capable of causing disease over very long time intervals. 

Introducing veterinary services, beekeepers, and beekeeping organizations to appropriate legal regulations, bee and brood diseases, GAP and GHP [6,8,65], proper beekeeping management, and HACCP systems is the basis for successful beekeeping. 

Dickel et al. conducted the first experimental trials of an oral vaccination against AFB disease, which represents a new milestone in the management of bee and other insect diseases [72].

The United State Department of Agriculture (USDA) has granted a two-year conditional license to a vaccine produced by Dalan Animal Health, an American biotech company specializing in immunology and insect health, that could help protect bees against the bacterial disease AFB [73]. 

The implementation of the mentioned measures requires the education of all participants in the process of beekeeping production.

## 7. Good Beekeeping Practice (GAP+GHP)—A Reminder for Beekeepers

Taking adequate measures in primary beekeeping production is the gold standard in the prevention of AFB and other disease of bee colonies [4,7,41,42,43,45,65,68,69]. This incolves:Meeting the legal requirements for beekeepingBeekeeping in adequate areas and appropriate terrainsBeekeeping is done in adequate hives that are disinfected prior to the bees’ settling inUsing a sterilized comb foundationA laboratory investigation of comb foundation for the presence of *P. larvae* endosporesRecord of work on the apiary (beekeeper’s diary)Beekeeping using strong colonies with high-quality food and good arrangementProviding adequate bee nests with good ventilationUsing young, fertile queens in beekeepingProviding water devices in the apiaryNot merging weak and strong colonies (sick and healthy)Regularly checking bee colonies and providing replacement combsBee colonies: compulsory checking (inspection) of bee colonies when bees are foragingImplementing proper disinfection during normal work at the apiaryUndertaking measures for early disease detection (regular monitoring)Control of stress factors in beekeepingPerforming laboratory diagnosticsBee colonies infected with the AFB pathogen agent are destroyed by burning and burying any infected bee colonyImplementing disinfection measures in the apiary in the case of occurrences of diseases

## Figures and Tables

**Figure 1 vetsci-10-00180-f001:**
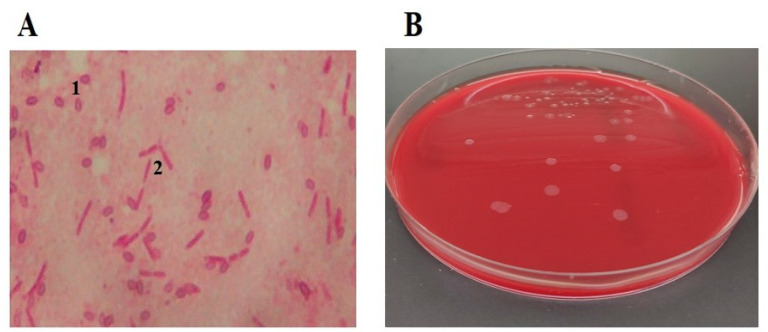
Microscopic view (1000×) of *P. larvae* endospores 1 and vegetative cells 2 (**A**) and *P. larvae* cultured on Columbia Blood Agar (**B**).

**Figure 2 vetsci-10-00180-f002:**
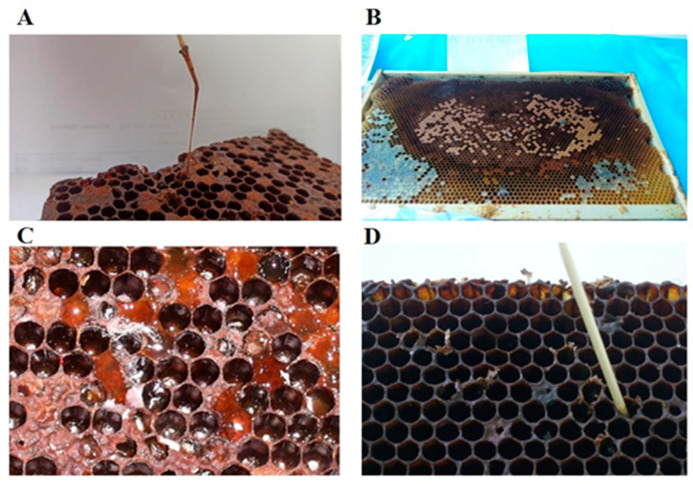
The dead larva in the bee brood cell transformed into ropy mass (**A**); Scattered bee brood cells—larvae infected with the causative agent AFB (**B**); Moisture, the change of color, concavity, and holes in cell caps bee brood (**C**); Bee comb cells after cleaning and desiccation of the contents of dead larvae (**D**).

**Figure 3 vetsci-10-00180-f003:**
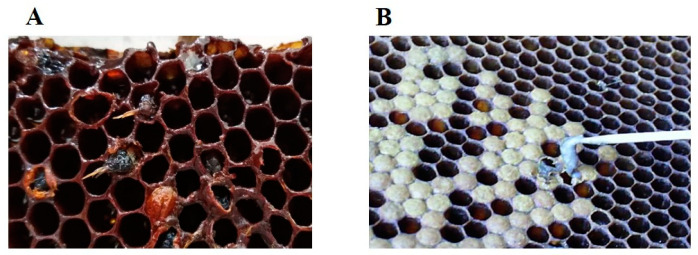
Projecting tongue of dead pupae, one of the rare signs of AFB (**A**) [30] and diseased larvae in a case of sacbrood disease (**B**).

**Figure 4 vetsci-10-00180-f004:**
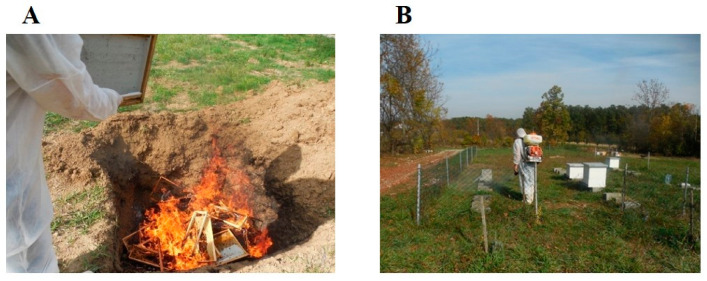
Burning and burying infected bee colonies after destroying the bees to control AFB (**A**) and disinfection of the apiary with a 10% Ca(OH)_2_ solution, in the evening hours, after stamping out AFB-infected bee colonies (**B**).

**Table 1 vetsci-10-00180-t001:** *P. larvae* genotypes and their characteristics.

Genotype	ERIC I	ERIC II	ERIC III	ERIC IV	ERIC V
Virulence	Kill larvae within 12 days	Kills larvae within 7 days	Kills larvae within 7 days		Kills even after 3 days
Frequency	Most frequent genotype, found throughout the world	Isolated worldwide, especially in Europe	Not identified in recent decades		Identified in Spain
References	[2,22,29]	[2,22,29]	[2,22,29]		[26]

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
