# Peer review of "American Foulbrood—Old and Always New Challenge"

_vetsci, 2023, doi:10.3390/vetsci10030180_

Round 1

Reviewer 1 Report (New Reviewer)

Thank you for the opportunity to consider the article titled, “American foulbrood - old and always new challenge” by Kazimir Matović, et al. This manuscript reviews an ongoing problem of American foulbrood infection in apiculture in regions where honey bees are kept. 

 In several locations in the manuscript, wording changes or other edits are suggested to improve content delivery.

1.    throughout the manuscript, the word “spore” should be replaced with “endospore”

2.    Figure 1A should contain a scale bar for reference, total magnification, and the type of stain used for visualization

3.    Figure 1B should be shown in higher magnification and increased resolution so that morphology of individual colonies can be observed

4.    line 64, the phrase, “in the brood that died larvae” is awkward

5.    line 78, the sentence beginning in this line suggests all five ERIC genotypes have different forms of the endospore, can this be explained in greater detail?

6.    line 89, replace word “proven” with “shown”

7.    line 101, the sentence beginning on this line is long and should be condensed or separated into more than one sentence

8.    line 107, there appears to be confusion with “infectious dose” and “lethal dose”, if assumed to be the same, this should be stated

9.    line 115 “have got to play” is awkward

10. line 120, “the large of the larvae” is awkward

11. line 129, replace “born” with “hatched”

12. line 133, “a lot virulence” is vague and should be reworded

13. line 143, P. larvae should be italicized

14. line 162, the sentence beginning in this line is long and awkward

15. line 168, sentence beginning, “Since ABF is a disease bee… is awkward, and insert “of” between “disease” and “bee”

16. line 194, the caption for Figure 2D is awkward

17. line 204, the sentence beginning near the end of this line is awkward and should be shortened

18. line 211, remove the word “sometimes”

19. line 219, GAP and GHP should be spelled out

20. line 279, the use of the word “anamnestic” seems unusual as it is used in this sentence

21. line 304, Figures 4 A and B do not seem informative (especially Figure 4B), and could be replaced with some reference for acceptable practice for packaging samples

22. line 308, the sentence beginning in this line is awkward and should be condensed

23. line 315, use of the phrase, “evil(use)” is not clear

24. line 317, the phrase, “will have an adverse effect on the saprophytic microflora of humans” does not seem justified especially because the choice for word, “saprophytic” does not appear to be correct

25. line 321, the phrase, “honey from the hives in diseased colonies” should be condensed

26. line 358, reference 25 does not appear to involve microbiome analysis

27. line 364, not all of these media appear to contain components (such as antibiotics) that make them selective, it may be more appropriate to refer to at least some of these as “rich” media

28. line 365, the name of MYPGP medium should be spelled out

29. line 372, this sentence is awkward and there appears to be a parenthetical marker missing

30. line 390, the word, “exactly” should be removed

31. line 420, the phrase “stamping out” here and in other lines is not needed and can be removed

32. line 423, the sentence beginning in this line is awkward and should be shortened

33. line 429, it is not clear how disinfection of the apiary is taking place in Figure 6B, and this figure should be removed

34. line 432, the acronym HACCP should be spelled out

35. the authors may be interested in including mention and citation of a recent publication on vaccination of queens to American Foulbrood disease (reference below)

Front Vet Sci, 2022 Oct 17;9:946237. doi: 10.3389/fvets.2022.946237. eCollection 2022

Title: The oral vaccination with Paenibacillus larvae bacterin can decrease susceptibility to American Foulbrood infection in honey bees-A safety and efficacy study

Authors: Franziska Dickel, Nick Maria Peter Bos, Huw Hughes, Raquel Martín-Hernández, Mariano Higes, Annette Kleiser, and Dalial Freitak

Author Response

We are very grateful for time reviewers invested in reviewing this paper (American foulbrood - old and always new challenge- vetsci-2175402) and constructive criticisms and corrections you recommended in order to improve our manuscript.

We took into account all suggestions, and we hope you will find that quality of manuscript is improved. In general, according to reviewers suggestions, particular mistake are corrected. Thus, we believe the manuscript is rearranged to meet requirements of reviewers. Specific corrections and comments are presented in the table below, and all changes are highlighted in the manuscript.

Reviewer 1

Thank you for the opportunity to consider the article titled, “American foulbrood - old and always new challenge” by Kazimir Matović, et al. This manuscript reviews an ongoing problem of American foulbrood infection in apiculture in regions where honey bees are kept.

 In several locations in the manuscript, wording changes or other edits are suggested to improve content delivery.

  1. throughout the manuscript, the word “spore” should be replaced with “endospore”

Corrected

  1. Figure 1A should contain a scale bar for reference, total magnification, and the type of stain used for visualization

Corrected

  1. Figure 1B should be shown in higher magnification and increased resolution so that morphology of individual colonies can be observed

Replaced

  1. line 64, the phrase, “in the brood that died larvae” is awkward

Corrected

  1. line 78, the sentence beginning in this line suggests all five ERIC genotypes have different forms of the endospore, can this be explained in greater detail?

Corrected, references added

  1. line 89, replace word “proven” with “shown”

Corrected

  1. line 101, the sentence beginning on this line is long and should be condensed or separated into more than one sentence

Corrected, split sentence

  1. line 107, there appears to be confusion with “infectious dose” and “lethal dose”, if assumed to be the same, this should be stated

Corrected

  1. line 115 “have got to play” is awkward

Corrected

  1. line 120, “the large of the larvae” is awkward

Corrected

  1. line 129, replace “born” with “hatched”

Corrected

  1. line 133, “a lot virulence” is vague and should be reworded

Corrected

  1. line 143, P. larvae should be italicized

Corrected

  1. line 162, the sentence beginning in this line is long and awkward

Corrected

  1. line 168, sentence beginning, “Since ABF is a disease bee… is awkward, and insert “of” between “disease” and “bee”

Corrected

  1. line 194, the caption for Figure 2D is awkward

Corrected

  1. line 204, the sentence beginning near the end of this line is awkward and should be shortened

Shortened

  1. line 211, remove the word “sometimes”

Removed

  1. line 219, GAP and GHP should be spelled out

Corrected

  1. line 279, the use of the word “anamnestic” seems unusual as it is used in this sentence

Removed

  1. line 304, Figures 4 A and B do not seem informative (especially Figure 4B), and could be replaced with some reference for acceptable practice for packaging samples

Corrected and removed

  1. line 308, the sentence beginning in this line is awkward and should be condensed

Corrected

  1. line 315, use of the phrase, “evil(use)” is not clear

Corrected

  1. line 317, the phrase, “will have an adverse effect on the saprophytic microflora of humans” does not seem justified especially because the choice for word, “saprophytic” does not appear to be correct

Removed

  1. line 321, the phrase, “honey from the hives in diseased colonies” should be condensed

Corrected

  1. line 358, reference 25 does not appear to involve microbiome analysis

Deleted

  1. line 364, not all of these media appear to contain components (such as antibiotics) that make them selective, it may be more appropriate to refer to at least some of these as “rich” media

Removed

  1. line 365, the name of MYPGP medium should be spelled out

Corrected

  1. line 372, this sentence is awkward and there appears to be a parenthetical marker missing

We hope we have corrected

  1. line 390, the word, “exactly” should be removed

Removed

  1. line 420, the phrase “stamping out” here and in other lines is not needed and can be removed

Removed twice

  1. line 423, the sentence beginning in this line is awkward and should be shortened

Corrected, split

  1. line 429, it is not clear how disinfection of the apiary is taking place in Figure 6B, and this figure should be removed

Corrected and agreed with another reviewer

  1. line 432, the acronym HACCP should be spelled out

Corrected, already mentioned in line 235

  1. the authors may be interested in including mention and citation of a recent publication on vaccination of queens to American Foulbrood disease (reference below)

Included and added reference, number 64

Front Vet Sci, 2022 Oct 17;9:946237. doi: 10.3389/fvets.2022.946237. eCollection 2022

Title: The oral vaccination with Paenibacillus larvae bacterin can decrease susceptibility to American Foulbrood infection in honey bees-A safety and efficacy study

Authors: Franziska Dickel, Nick Maria Peter Bos, Huw Hughes, Raquel Martín-Hernández, Mariano Higes, Annette Kleiser, and Dalial Freitak

Reviewer 2 Report (New Reviewer)

This review focuses on the American Foulbrood (AFB), a highly contagious disease of the Western honeybee Apis mellifera. It describes the causative agent Paenibacillus larvae and its five different types, the disease’s development in the bee larvae and the mechanisms of its spread in the bee colony. Additionally the clinical signs and visual cues in the colony are discussed. AFB control including hygienic measurements, colony eradication and hive material cleaning are being reviewed, as well as methods for AFB monitoring and AFB diagnosis.

AFB is an important honeybee disease, which is notifiable in some regions such as the European Union. Informative and well-structured reviews can help to control and fight outbreaks of AFB, thus the authors’ endeavour is a useful one. However, in the recent form the review has too many flaws to be useful. It rather leads to confusion and disinformation. There has been put a lot of work into this review already, a fundamental revision may lead to a valuable contribution to this field of work. In this sense, my comments below should help to reach this goal.

Generally, the writing is poor and has to be revised thoroughly. The most evident problems are a flawed syntax and a careless punctuation. I suggest avoiding long and interlaced sentences, as short sentences are both easier to write and to understand. Furthermore, avoid to provide essential information inside parentheses.

The abstract content is rather vague. Please give an overview about the topics/ the focus of the review.

The review has to be structured more carefully. Some topics are mentioned in details over and over again without gaining any new information (e.g. ERIC types and their differences, hygienic behaviour of nurse bees). Other important topics are dealt with too briefly and are thus not understandable (e.g. the different ways of monitoring for AFB including the different matrixes and methods; the importance and management of subclinical infection; the management of the disease on colony level vs. apiary level).

Furthermore, the chapter structure has to be improved. The chapter ,AFB control’, for example, contains also information about AFB diagnosis techniques – these may fit with the ‘differential diagnosis’ chapter. Legal issues are mentioned throughout the text in a repetitive pattern, maybe an own chapter would be advisable. A short summary with your take-home messages would also be helpful for the reader.

Be careful with the legal aspect of the disease. The laws differ strongly between different regions regarding AFB. Either state clearly somewhere quite early, that your legal information concerns a special region or describe the different situations in different regions of the world. For this it may also be helpful to refer to the respective laws, that people can check quickly, whether these still apply. Differentiate between legal issues and good beekeeping practices during your writing, as these differ strongly in the consequences if ignored. You state, that antibiotics are only permitted in the US – I cannot believe that antibiotics are prohibited in all other countries in the world. Please check and change your phrasing.

Use technical terms more carefully (e.g brood cells are in a brood comb, not in a honey comb; a late symptom of AFB is a scale, not a scab; a cell cap cannot be diseased).

The photos in Fig. 3 A and B show very untypical symptoms. Authors should double-check if they really show the assigned disease!

The photos in Fig. 4 A and B are not convincing, as in picture A also brown spots are visible on the surface of the packing, so there has been leaking out. A description of the material (pieces of a brood comb) is also lacking.

The photo in Fig. 6 B needs some explanation of the measures taken: Which disinfectant was used? Why are there remaining colonies in the apiary? Note: this should be Fig 5. instead of 6.

List in the chapter ‘AFB prevention’: eventually convert in a table adding a column with categories, which divide the given measurments, e.g. general good beekeeping measurements, general disease prevention measurements, prevention measurements against AFG and control measurement against AFB . Another column can be added matching the measurements with the connected citations.

Author contributions: these categories make no sense, as no study is conceived or designed and no lab work was necessary.

Author Response

We are very grateful for time reviewers invested in reviewing this paper (American foulbrood - old and always new challenge- vetsci-2175402) and constructive criticisms and corrections you recommended in order to improve our manuscript.

We took into account all suggestions, and we hope you will find that quality of the manuscript is improved. In general, according to reviewers' suggestions, particular mistakes are corrected. Thus, we believe the manuscript is rearranged to meet the requirements of reviewers. Specific corrections and comments are presented in the table below, and all changes are highlighted in the manuscript.

This review focuses on the American Foulbrood (AFB), a highly contagious disease of the Western honeybee Apis mellifera. It describes the causative agent Paenibacillus larvae and its five different types, the disease’s development in the bee larvae and the mechanisms of its spread in the bee colony. Additionally the clinical signs and visual cues in the colony are discussed. AFB control including hygienic measurements, colony eradication and hive material cleaning are being reviewed, as well as methods for AFB monitoring and AFB diagnosis.

AFB is an important honeybee disease, which is notifiable in some regions such as the European Union. Informative and well-structured reviews can help to control and fight outbreaks of AFB, thus the authors’ endeavour is a useful one. However, in the recent form the review has too many flaws to be useful. It rather leads to confusion and disinformation. There has been put a lot of work into this review already, a fundamental revision may lead to a valuable contribution to this field of work. In this sense, my comments below should help to reach this goal.

Generally, the writing is poor and has to be revised thoroughly. The most evident problems are a flawed syntax and a careless punctuation. I suggest avoiding long and interlaced sentences, as short sentences are both easier to write and to understand. Furthermore, avoid to provide essential information inside parentheses.

Corrected

The abstract content is rather vague. Please give an overview about the topics/ the focus of the review.

Corrected

The review has to be structured more carefully. Some topics are mentioned in details over and over again without gaining any new information (e.g. ERIC types and their differences, hygienic behaviour of nurse bees). Other important topics are dealt with too briefly and are thus not understandable (e.g. the different ways of monitoring for AFB including the different matrixes and methods; the importance and management of subclinical infection; the management of the disease on colony level vs. apiary level).

Corrected

Furthermore, the chapter structure has to be improved. The chapter ,AFB control’, for example, contains also information about AFB diagnosis techniques – these may fit with the ‘differential diagnosis’ chapter. Legal issues are mentioned throughout the text in a repetitive pattern, maybe an own chapter would be advisable. A short summary with your take-home messages would also be helpful for the reader.

Rearranged, found compromise solution between reviewers

Be careful with the legal aspect of the disease. The laws differ strongly between different regions regarding AFB. Either state clearly somewhere quite early, that your legal information concerns a special region or describe the different situations in different regions of the world. For this it may also be helpful to refer to the respective laws, that people can check quickly, whether these still apply. Differentiate between legal issues and good beekeeping practices during your writing, as these differ strongly in the consequences if ignored. You state, that antibiotics are only permitted in the US – I cannot believe that antibiotics are prohibited in all other countries in the world. Please check and change your phrasing.

Corrected, added reference

Use technical terms more carefully (e.g brood cells are in a brood comb, not in a honey comb; a late symptom of AFB is a scale, not a scab; a cell cap cannot be diseased).

Corrected

The photos in Fig. 3 A and B show very untypical symptoms. Authors should double-check if they really show the assigned disease!

Yes, in the laboratory of VSI Kraljevo, in the mentioned sample (3A - bee brood and pupae), the presence of endospores and the genome of P. larvae was confirmed microscopically and by PCR. larvae. Also in the sample from Figure 3B, in the bee brood/larvae, the presence of sacbrood virus was determined by the Real-Time PCR method.

The photos in Fig. 4 A and B are not convincing, as in picture A also brown spots are visible on the surface of the packing, so there has been leaking out. A description of the material (pieces of a brood comb) is also lacking.

Deleted and corrected

The photo in Fig. 6 B needs some explanation of the measures taken: Which disinfectant was used? Why are there remaining colonies in the apiary? Note: this should be Fig 5. instead of 6.

Corrected; healthy, clinical signs disease – free hives remained in the apiary

List in the chapter ‘AFB prevention’: eventually convert in a table adding a column with categories, which divide the given measurments, e.g. general good beekeeping measurements, general disease prevention measurements, prevention measurements against AFG and control measurement against AFB . Another column can be added matching the measurements with the connected citations.

Rearranged, found compromise solution between reviewers

Author contributions: these categories make no sense, as no study is conceived or designed and no lab work was necessary.

Deleted

Reviewer 3 Report (New Reviewer)

The comments are attached as separate file. 

The detailed comments are in the manuscript file

Author Response

There are specific questions/ amendments proposed in the review. These can be can be found in the manuscript.

General comments

  • Check and italicize the S, Name
  • Add the description of abbreviations at its first place of use
  • Write complete name of genus at its first place of use

            Corrected

Abstract:

  • What is the outcome of this review? Abstract gave general information but did not highlight the findings/ outcome of the current review.

Corrected

  • Introduction:

o          Any worldwide estimation for damage caused by AFB?

            Corrected, added text and references 20 and 21

o          Are the figures by the author or adopted from any previous study. If later is the case, add the reference

o          Table 1: Add list of references in the last column of the table for each genotype.

            Added

  • Subheadings for different controlling methods could improve the readability.

Corrected

o          The review should include the data about commonly used antibiotics against AFB?

            It already has lines 318-319

  • Line 373: NO comparison is included in the earlier sections of review between AFB and EFB. Add the causal organism of EFB.

YES, compared later, in Differential diagnostics of the disease, line . Added

o          Line 343-349: Any references to support these facts?

Added references

o          Line 405-422: What are these points? Add suitable sentence/ subheading to introduce these points? Proposed Precautionary measures etc. What are these points? Add suitable sentence/ subheading to introduce these points? Proposed Precautionary measures etc

Rearranged, found compromise solution between reviewers

o          The author should include the summary/ conclusion of this review. How this review   will increase the knowledge about AFB and what will be the potential use of this information by the beekeepers.  The author must establish the strong connection that how this review regarding AFB will be helpful for the beekeepers and Apiculture industry.

Included

References

  • Italicize the S. names
  • Cross check the references in the text
  • Check the recommended format of the journal for writing the journal names.

Corrected

Reviewer 4 Report (New Reviewer)

Comment and suggestion to Authors:

I find this paper very well written; the manuscript is a useful short overview of what is known about AFB.

Here some minor revision to apply before publication:

Line 28: Pestisapium to Pestis apium

Line 45: rephrase the sentence

Line 46: I suggest to rephrase in “Except Morrisey et al. 2014, little officially data is available on the prevalence of AFB…”

Line 64: remove the extra”]”

Line 143: P. larvae in P. larvae

Line 374: missing “)”

Author Response

We are very grateful for the time reviewers invested in reviewing this paper (American foulbrood - old and always new challenge- vetsci-2175402) and constructive criticisms and corrections you recommended in order to improve our manuscript.

We took into account all suggestions, and we hope you will find that quality of the manuscript is improved. In general, according to reviewers' suggestions, particular mistakes are corrected. Thus, we believe the manuscript is rearranged to meet the requirements of reviewers. Specific corrections and comments are presented in the table below, and all changes are highlighted in the manuscript.

Comment and suggestion to Authors:

I find this paper very well written; the manuscript is a useful short overview of what is known about AFB.

Here some minor revision to apply before publication:

Line 28: Pestisapium to Pestis apium

Corrected

Line 45: rephrase the sentence

Rephrased the sentence

Line 46: I suggest to rephrase in “Except Morrisey et al. 2014, little officially data is available on the prevalence of AFB…”

Corrected

Line 64: remove the extra”]”

Removed

Line 143: P. larvae in P. larvae

Corrected

Line 374: missing “)”

Added

We would love to thank you for allowing us to resubmit a revised copy of the manuscript. We believe that the revised paper is more comprehensive, and we do hope that it can meet your expectation.

Sincerely,

Jelena Ćirić, DVM, PhD

Added four references numbers: 20, 21, 64 and 71

Document selectedA8-0014/2018 ... OPINION of the Committee on the EnvironmentPublic Health and Food Safety, 2018.

Smith MR, Mueller ND, Springmann M, Sulser TB, Garibaldi LA, Gerber J, Wiebe K, Myers SS. Pollinator Deficits, Food Consumption, and Consequences for Human Health: A Modeling Study. Environ Health Perspect. 2022 Dec; 130(12):127003. doi: 10.1289/EHP10947. Epub 2022 Dec 14. PMID: 36515549; PMCID: PMC9749483.

ANONYMOUS (2022): OIE Terrestrial Animal Health Code. Apidae. Terrestrial Code Online Access. Section 9. Chapter 9.2.

Dickel F, Bos NMP, Hughes H, Martín-Hernández R, Higes M, Kleiser A, Freitak D. The oral vaccination with Paenibacillus larvae bacterin can decrease susceptibility to American Foulbrood infection in honey bees-A safety and efficacy study. Front Vet Sci. 2022 Oct 17;9:946237. doi: 10.3389/fvets.2022.946237. PMID: 36325099; PMCID: PMC9618583

Round 2

Reviewer 3 Report (New Reviewer)

The author has significantly improved the manuscript.  However, spelling check is still recommended for the manuscript

Author Response

Dear Reviewer,

We are very grateful for the time reviewer invested in reviewing this paper (American foulbrood - old and always new challenge- vetsci-2175402) and the constructive criticisms and corrections you recommended in order to improve our manuscript.

We took into account all suggestions, and we hope you will find that quality of the manuscript is improved. In general, according to reviewers' suggestions, particular mistakes are corrected. Thus, we believe the manuscript is rearranged to meet the requirements of reviewers. All changes are highlighted in the manuscript.

This manuscript is a resubmission of an earlier submission. The following is a list of the peer review reports and author responses from that submission.

Round 1

Reviewer 1 Report

The review article "American foulbrood - old and always new challenge" contains a description of the infectious disease American foulbrood (AFB) and its pathogen Paenibacillus larvae (P. larvae). The diagnosis of this disease is described and possible preventive measures that the beekeeper can use are explained in detail.

basic comments

Already on the first reading it becomes clear that many of the points raised are described far too inaccurately and insufficiently explained. This can lead to misunderstandings and misjudgments on the part of the reader. Other sections, on the other hand, are then described in great detail, so it is not entirely clear to whom the review is addressed.

specific comments

1. Introduction

Line 40-41

It is stated that only few data on the prevalence of AFB are available to date. The MLST data known since 2014 (Morrissey et al., 2014) were not considered here. However, samples collected worldwide were tested for P. larvae by Morrissey and colleagues.

2. American foulbrood and the etiology of the causative agent of the disease

Line 51

It is described that P. larvae was previously classified as Bacillus larvae. That’s correct. The intermediate step of the classification to Paenibacillus larvae subsp. larvae and Paenibacillus larvae subsp. pulvifaciens is not described, although this is even mentioned in Image 1. Thus, this was not sufficiently explained.

Line 69-70

The authors describe results that are obtained. However, it is not clear what the specific results are. Does this mean the detection of P. larvae or the genotyping of the bacteria? The connection is not clear. Djukic et al. (2014) is given as a reference. In this paper, the sequencing of the entire genome of two P. larvae strains and the subsequent annotation of the genomes is described and the results are explained and discussed. The detection of P. larvae is not an issue here.

Line 70-73

The two successive sentences have the same meaning.

Line 77-78

The authors explain that some P. larvae genotypes cause the same symptoms. Only some? What are the symptoms in strains of the other genotypes? Just because the beekeeper or veterinarian sees some symptoms rarely does not mean that they do not occur.

Line 79-80

It is correctly described that the P. larvae genotypes differ, among other things, in their metabolism (carbohydrate turnover). An essential reference for this is that of Neuendorf et al. (2004). This should be added.

Image 1

Why is an image inserted here and not a proper table? Copying a table completely from another manuscript and not filling a separate table with corresponding information is not state of the art.

In addition, P. larvae genotypes III and IV are classified into the subspecies pulvifaciens. The classification of P. larvae into subspecies was reclassified in 1996 (Genersch et al., 1996). All genotypes thus belong to P. larvae. Furthermore, as with the other genotypes, the LT100 should be specified for ERIC II.

Line 88-92

A bee colony infected with P. larvae ERIC II may be misdiagnosed as P. larvae negative when examined for typical AFB symptoms. Why is this so? This is well known and needs to be explained in more detail here.

False negative diagnoses due to aqueous components of the larvae (lines 91-92, 162-163) are unknown to me. What does that mean? This statement is incomprehensible. The reader can only speculate. In addition, an incorrect reference has been given here as well. The cited review by Genersch et al. (2010) does not describe this misjudgment of symptoms.

Line 93-95, 99-100

Authors need to express themselves more clearly, so that their presumably correct statements are also correct. For example, dead larvae are removed from an infected colony by worker bees. They then try to clean the corresponding brood cells again. P. larvae spores can attach themselves to the hair coat of the bees, to the mandibles and extremities. Through their normal activities in the bee colony, these bees then distribute the spores in the colony and also the larval food can be contaminated. A new infection of larvae then takes place again via the food. An infection does not develop from the larval food or from the uncapped brood, as described here. The larval food, for example, would have to be already contaminated.

Line 101-102

A reference to this statement is completely missing. The American Foulbrood is a brood disease. Adult bees are not susceptible to this disease.

Line 102-105

Hygienic bees can clear a large part of the diseased brood before capping. Of course, this also removes bacteria and spores from the colony. Then the larvae are often still largely intact on the outside. The spores are distributed to a lesser extent in the colony, since the bees do not necessarily come into direct contact with them. How then can these bees play an important role in the spread of spores? This needs to be explained in detail.

Line 108-110

A reference to this statement is completely missing.

Line 111

A wrong word has been used here, which makes the statement of the sentence inaccurate. P. larvae do not produce more virulence factors. If that were the case, a comparison parameter would have to be included. P. larvae produce many virulence factors.

Line 118-120

Likewise, here too a statement is presented completely incorrectly, due to the wrong choice of words. Bees are not pathogens. They only can transmit the spores of the pathogen.

Line 135-136

Why is this example given here? It is not typical for P. larvae to also infect humans. Misuse of drugs, substances, equipment, chemicals ect. can always lead to undesirable consequences. If such an example is to be given, it must at least be explained more precisely and viewed in a very differentiated way.

In addition, no honeybee was injected in this case.

3. Clinical signs of AFB

Line 142-143

Why should changes of the larva (colour, consistency, etc. ) not be preserved? These are not actually short-term effects in a P. larvae infection.

Line 146-147

It is described here that changes in the cell cover occur after 20 days post-infection. At this point, an infected larva is long dead. How can this be explained?

Line 170-173

The ERIC I genotype is not more common than the ERIC II genotype (see MLST by Morrissey et al.). The symptoms are just easier to see. In addition, when larva infected with P. larvae ERIC I, the larvae do not only die after capping. They also die in the uncapped stage. Only fewer larvae die at this point when infected with P. larvae ERIC II. It is important to represent such points correctly.

Line 174-176, Figure 3A

Protrusion of the tongue is, if at all, a very rare symptom of infection with P. larvae. Please provide the appropriate reference. The reference given here does not describe or explain this. Bees that die from other causes after hatching can also stick out their tongue. The bees in the figure may also have starved after hatching or died from hypothermia. They obviously nibbled through the cell lid after hatching. Pupae that die from P. larvae infection die under the closed cell lid. How did this picture come about? Explanations of this would be helpful.

Figure 3B

An image of a sacbrood infection is shown. This is not further explained.

4. AFB control

This chapter describes a number of measures and applications to be carried out in the event of an AFB infection. These requirements may differ in different countries, which must also be taken into account in this review. Such general statements cannot be made.

Line 213-215

In order to detect a P. larvae infection, both the positive laboratory evidence and typical AFB symptoms in the colony must be demonstrated. If one point cannot be achieved, the colony is declared uninfected. But that doesn’t mean that an infection has ended, it means that there is no infection.

Line 222-223

It is possible, as described here, that AFB symptoms are first recognized and then the pathogen is detected in the laboratory. However, in a preventive check-up, it takes place exactly the other way around. First the pathogen is detected in the laboratory and then the colony is examined for possible symptoms. Both possibilities need to be explained and discussed here. Generalized in this way, the statement cannot stand.

Line 255-261, 264-268

This section is completely redundant. Samples must be sent according to the instructions of the testing laboratory.

Line 269-273

The use of antibiotics is not prohibited everywhere. This aspect needs to be considered in a more differentiated way. There are, of course, a number of reasons against the use of antibiotics. However, there is no general ban (see USA). Such facts must be explained and discussed in a review.

Line 273-276

The honey is marketable and not unusable. However, the rules may vary from country to country.

Figure 5

This figure is not noted and described in the text. The used marker is unsuitable because the uppermost band is smaller than the PCR product shown. The PCR is not described at all. Which PCR was performed? Which primers were used? What has been proven? Is the amplicon P. larvae specific?

In general, only published results are summarized and evaluated in a review article. Own experimental results are not published.

5. AFB differential diagnostics

Line 284-285

There is a possibility of confusion with other brood diseases during the diagnosis. However, this can only happen at first glance. This is not possible with an exact laboratory examination. However, it is not clear to me how AFB can be confused with a varroosis. If such facts are asserted, they must also be explained and substantiated. The reference given does not provide any information on sackbrood or varroosis.

Line 309-311

Various methods for detecting P. larvae are listed. Only one reference of 2021 is given for this. These methods have been established for decades. It is not sufficient to provide a reference in which only one of these methods is used. The corresponding original papers must be cited. What method is available to detect P. larvae based on antibodies?

Line 315-318

It is correct that an endolysin CBD that binds P. larvae has been identified. This domain has the potential to be used for the detection of P. larvae. As far as I know, such a test does not yet exist. Please provide the appropriate reference. Reference 55 does not describe this.

6. AFB prevention

Line 347-349

The existence of an association of beekeepers does not automatically produce healthy colonies.

Line 360

The larvae spores described here do not exist. Probably P. larvae spores are meant.

Line 381

What is a HACCP system? The explanation for this is missing. In general, words that are abbreviated should be spelled out in full at least once for explanation.

7. References

Other reviews are often used as references for this review article. It would be better to indicate the original articles. Then it is easier for the reader to find out and understand more quickly how certain facts were experimentally proven.

Author Response

We are very grateful for time reviewers invested in reviewing this paper (American foulbrood - old and always new challenge- vetsci-1999546) and constructive criticisms and corrections you recommended in order to improve our manuscript.

We took into account all suggestions, and we hope you will find that quality of manuscript is improved. In general, according to reviewers suggestions, particular mistake are corrected. Thus, we believe the manuscript is rearranged to meet requirements of reviewers. Specific corrections and comments are presented in the table below, and all changes are highlighted in the manuscript.

Reviewer 1

The review article "American foulbrood - old and always new challenge" contains a description of the infectious disease American foulbrood (AFB) and its pathogen Paenibacillus larvae (P. larvae). The diagnosis of this disease is described and possible preventive measures that the beekeeper can use are explained in detail.

-Thank you for these comments.

Basic comments

Already on the first reading it becomes clear that many of the points raised are described far too inaccurately and insufficiently explained. This can lead to misunderstandings and misjudgments on the part of the reader. Other sections, on the other hand, are then described in great detail, so it is not entirely clear to whom the review is addressed.

  • Thank you for these comments.

specific comments

  1. Introduction

Line 40-41

It is stated that only few data on the prevalence of AFB are available to date. The MLST data known since 2014 (Morrissey et al., 2014) were not considered here. However, samples collected worldwide were tested for P. larvae by Morrissey and colleagues.

- Thank you for these comments, this reference is on 23. position.

  1. American foulbrood and the etiology of the causative agent of the disease

Line 51

It is described that P. larvae was previously classified as Bacillus larvae. That’s correct. The intermediate step of the classification to Paenibacillus larvae subsp. larvae and Paenibacillus larvae subsp. pulvifaciens is not described, although this is even mentioned in Image 1. Thus, this was not sufficiently explained.

  • Thank you for these comments, the image is replaced as table and its corrected in the main text.

Line 69-70

The authors describe results that are obtained. However, it is not clear what the specific results are. Does this mean the detection of P. larvae or the genotyping of the bacteria? The connection is not clear. Djukic et al. (2014) is given as a reference. In this paper, the sequencing of the entire genome of two P. larvae strains and the subsequent annotation of the genomes is described and the results are explained and discussed. The detection of P. larvae is not an issue here.

  • Yes, its deleted (reference 22, 26)

Line 70-73

The two successive sentences have the same meaning.

  • One of sentences is deleted.

Line 77-78

The authors explain that some P. larvae genotypes cause the same symptoms. Only some? What are the symptoms in strains of the other genotypes? Just because the beekeeper or veterinarian sees some symptoms rarely does not mean that they do not occur.

  • Yes, this is corrected in the text

Line 79-80

It is correctly described that the P. larvae genotypes differ, among other things, in their metabolism (carbohydrate turnover). An essential reference for this is that of Neuendorf et al. (2004). This should be added.

  • The reference of Neuendorf et al. (2004). is added.

Image 1

Why is an image inserted here and not a proper table? Copying a table completely from another manuscript and not filling a separate table with corresponding information is not state of the art.

  • The image is repleced as table.

In addition, P. larvae genotypes III and IV are classified into the subspecies pulvifaciens. The classification of P. larvae into subspecies was reclassified in 1996 (Genersch et al., 1996). All genotypes thus belong to P. larvae. Furthermore, as with the other genotypes, the LT100 should be specified for ERIC II.

  • Yes, this is added.

Line 88-92

A bee colony infected with P. larvae ERIC II may be misdiagnosed as P. larvae negative when examined for typical AFB symptoms. Why is this so? This is well known and needs to be explained in more detail here.

  • Yes, this is added.

False negative diagnoses due to aqueous components of the larvae (lines 91-92, 162-163) are unknown to me. What does that mean? This statement is incomprehensible. The reader can only speculate. In addition, an incorrect reference has been given here as well. The cited review by Genersch et al. (2010) does not describe this misjudgment of symptoms.

  • This is removed from text.

Line 93-95, 99-100

Authors need to express themselves more clearly, so that their presumably correct statements are also correct. For example, dead larvae are removed from an infected colony by worker bees. They then try to clean the corresponding brood cells again. P. larvae spores can attach themselves to the hair coat of the bees, to the mandibles and extremities. Through their normal activities in the bee colony, these bees then distribute the spores in the colony and also the larval food can be contaminated. A new infection of larvae then takes place again via the food. An infection does not develop from the larval food or from the uncapped brood, as described here. The larval food, for example, would have to be already contaminated.

- Yes, the authors expressed themselves more clearly

Line 101-102

A reference to this statement is completely missing. The American Foulbrood is a brood disease. Adult bees are not susceptible to this disease.

  • Thank you for this comment.

Line 102-105

Hygienic bees can clear a large part of the diseased brood before capping. Of course, this also removes bacteria and spores from the colony. Then the larvae are often still largely intact on the outside. The spores are distributed to a lesser extent in the colony, since the bees do not necessarily come into direct contact with them. How then can these bees play an important role in the spread of spores? This needs to be explained in detail.

  • Clarified, new reference added

Fries I, Camazine S. Implications of horizontal and vertical pathogen transmission for honey bee epidemiology. Apidologie. 2001;32:199–214. https://doi.org/10.1051/apido:2001122.  

Line 108-110

A reference to this statement is completely missing.

  • This is added.

Line 111

A wrong word has been used here, which makes the statement of the sentence inaccurate. P. larvae do not produce more virulence factors. If that were the case, a comparison parameter would have to be included. P. larvae produce many virulence factors.

  • Yes, this is added.

Line 118-120

Likewise, here too a statement is presented completely incorrectly, due to the wrong choice of words. Bees are not pathogens. They only can transmit the spores of the pathogen.

  • Yes, this is added.

Line 135-136

Why is this example given here? It is not typical for P. larvae to also infect humans. Misuse of drugs, substances, equipment, chemicals ect. can always lead to undesirable consequences. If such an example is to be given, it must at least be explained more precisely and viewed in a very differentiated way.

In addition, no honeybee was injected in this case.

  • This is removed from text.

  1. Clinical signs of AFB

Line 142-143

Why should changes of the larva (colour, consistency, etc. ) not be preserved? These are not actually short-term effects in a P. larvae infection.

  • C

Line 146-147

It is described here that changes in the cell cover occur after 20 days post-infection. At this point, an infected larva is long dead. How can this be explained?

  • We tried to explain in the main text.

Line 170-173

The ERIC I genotype is not more common than the ERIC II genotype (see MLST by Morrissey et al.). The symptoms are just easier to see. In addition, when larva infected with P. larvae ERIC I, the larvae do not only die after capping. They also die in the uncapped stage. Only fewer larvae die at this point when infected with P. larvae ERIC II. It is important to represent such points correctly.

  • Corrected, added reference 28.

Line 174-176, Figure 3A

Protrusion of the tongue is, if at all, a very rare symptom of infection with P. larvae. Please provide the appropriate reference. The reference given here does not describe or explain this. Bees that die from other causes after hatching can also stick out their tongue. The bees in the figure may also have starved after hatching or died from hypothermia. They obviously nibbled through the cell lid after hatching. Pupae that die from P. larvae infection die under the closed cell lid. How did this picture come about? Explanations of this would be helpful.

Corrected and added reference. From these pupae and larvae, on the same frame and in the same colony, AFB was microscopically diagnosed in our laboratory.

Figure 3B

An image of a sacbrood infection is shown. This is not further explained.

- This is added.

  1. AFB control

This chapter describes a number of measures and applications to be carried out in the event of an AFB infection. These requirements may differ in different countries, which must also be taken into account in this review. Such general statements cannot be made.

  • Yes, this is in general.

Line 213-215

In order to detect a P. larvae infection, both the positive laboratory evidence and typical AFB symptoms in the colony must be demonstrated. If one point cannot be achieved, the colony is declared uninfected. But that doesn’t mean that an infection has ended, it means that there is no infection.

  • Yes!

Line 222-223

It is possible, as described here, that AFB symptoms are first recognized and then the pathogen is detected in the laboratory. However, in a preventive check-up, it takes place exactly the other way around. First the pathogen is detected in the laboratory and then the colony is examined for possible symptoms. Both possibilities need to be explained and discussed here. Generalized in this way, the statement cannot stand.

  • Corrected

Line 255-261, 264-268

This section is completely redundant. Samples must be sent according to the instructions of the testing laboratory.

  • This part is removed from text.

Line 269-273

The use of antibiotics is not prohibited everywhere. This aspect needs to be considered in a more differentiated way. There are, of course, a number of reasons against the use of antibiotics. However, there is no general ban (see USA). Such facts must be explained and discussed in a review.

  • Yes, this is added in the text.

Line 273-276

The honey is marketable and not unusable. However, the rules may vary from country to country.

  • Yes!

Figure 5

This figure is not noted and described in the text. The used marker is unsuitable because the uppermost band is smaller than the PCR product shown. The PCR is not described at all. Which PCR was performed? Which primers were used? What has been proven? Is the amplicon P. larvae specific?

- Described. The figure legend has also been corrected. I agree with you that the marker used is inappropriate because the top band is smaller than the PCR product shown, we didn't have another marker then. Given that the amplicons of the tested samples matched the reference material, we decided to display the image.

In general, only published results are summarized and evaluated in a review article. Own experimental results are not published.

  • Thank you for this point.

  1. AFB differential diagnostics

Line 284-285

There is a possibility of confusion with other brood diseases during the diagnosis. However, this can only happen at first glance. This is not possible with an exact laboratory examination. However, it is not clear to me how AFB can be confused with a varroosis. If such facts are asserted, they must also be explained and substantiated. The reference given does not provide any information on sackbrood or varroosis.

  • Corrected

Line 309-311

Various methods for detecting P. larvae are listed. Only one reference of 2021 is given for this. These methods have been established for decades. It is not sufficient to provide a reference in which only one of these methods is used. The corresponding original papers must be cited. What method is available to detect P. larvae based on antibodies?

  • Corrected

Line 315-318

It is correct that an endolysin CBD that binds P. larvae has been identified. This domain has the potential to be used for the detection of P. larvae. As far as I know, such a test does not yet exist. Please provide the appropriate reference. Reference 55 does not describe this.

  • Yes, corrected

  1. AFB prevention

Line 347-349

The existence of an association of beekeepers does not automatically produce healthy colonies.

Line 360

The larvae spores described here do not exist. Probably P. larvae spores are meant.

  • Yes, its true.

Line 381

What is a HACCP system? The explanation for this is missing. In general, words that are abbreviated should be spelled out in full at least once for explanation.

- Yes this is added in the text.

  1. References

Other reviews are often used as references for this review article. It would be better to indicate the original articles. Then it is easier for the reader to find out and understand more quickly how certain facts were experimentally proven.

-We try to use novel references in this scientific field.

Reviewer 2 Report

It would be good to comment on the possible presence or absence of different levels of resistance to AFB in the colony in relation to clinical signs of disease
Comment on the experience of your work in the diagnosis and/or eradication is welcome. 

Author Response

We are very grateful for time reviewers invested in reviewing this paper (American foulbrood - old and always new challenge- vetsci-1999546) and constructive criticisms and corrections you recommended in order to improve our manuscript.

We took into account all suggestions, and we hope you will find that quality of manuscript is improved. In general, according to reviewers suggestions, particular mistake are corrected. Thus, we believe the manuscript is rearranged to meet requirements of reviewers. Specific corrections and comments are presented in the table below, and all changes are highlighted in the manuscript.

Reviewer 2

It would be good to comment on the possible presence or absence of different levels of resistance to AFB in the colony in relation to clinical signs of disease
Comment on the experience of your work in the diagnosis and/or eradication is welcome. 

Respected,

Until now, in the Republic of Serbia, we have not had monitoring to determine the presence of spores of P. larvae neither as regular nor as determining the presence of spores in apiaries where the presence of the disease was confirmed, in communities that did not have a clinical picture of the disease, so this time we cannot declare the levels of resistance in relation to the clinical signs of the disease.

Thank you for this comment.

Reviewer 3 Report

The article is very interesting and gathers important information on current knowledge of such important disease. However, some work needs to be done on the language and citation of appropriate scientific papers. Moreover, the article could benefit from more critical assessment of where the research in the field is going in terms of prevention of infection.  More detailed review is as follows:

Maybe the abstract needs to refer more to the entire take home message of the paper if more is added on critical assessment of novel steps to prevent the disease. 

Line 34 – add a reference for the OIE.

Line 41. Space missing after the references

Line 44-48. Consider making this 2 phrases instead of one. It is long and disconnected. 

Line 62 – 63: use parenthesis instead of [] and throughout the text.

Is there any specific selective media for P. larvae? I am assuming that the blood agar is not specific for the growth of P. larvae.

Line 69-70: Results of what?

Line 91-92: what is the gold standard technique for diagnosis of foulbrood? I think the paper would be more complete if some attention is given to the limitations of diagnosis and improvements needed. For example, should the gold standard be culture-dependent combined with further molecular techniques to determine genotypes? Is there any specific selective media for P. larvae? I am assuming that the blood agar is not specific for the growth of P. larvae.

Line 102-105. Please, rephrase. This statement is long, confusing, and not scientific. 

Line 107-110. Same here, please rephrase and cite references to back up these statements. 

Line 118. Missing a space

Line 119. The bees are not causative agents, they might be called vectors. 

Line 128. Replace “get the disease” with scientific language.

Line 136. Please revise this sentence. They injected honey bee?

Line 140. Space missing

Line 142-164. Please, revise language. Here again, language is not scientific and references are missing to support content. 

Line 189-197: What are the early signs they are looking for? Are there any surveillance sample collection and testing being done? What is the recommendation?

Line 221. Maybe expand on the special equipment. It is not clear the importance of this phrase here.

Line 223. Double “and” and space missing

Line 237. space missing

Do you have references for your sampling instructions ?

Line 269-273. The development of antimicrobial resistance and the risk of antimicrobials in the honey is a VERY important part of the story. Maybe expanding on it would be ideal. This phrase is very long and should be expanded to reflect the importance of the issue. 

Figure 5 is not mentioned in the text. More information on the PCR diagnostic tool is needed. For example, what is the gene that the PCR is amplifying? Also, I don’t understand the legend. Is the 3737 a strain?

Line 285 and 289. Missing spaces

First 3 paragraphs of the differential diagnosis: here again, more references and careful language is needed. 

Interchangeable use of honeybee and honey bee. Stick to one format. 

AFB prevention: is the disease notifiable? Is data on the prevalence available? It would be a good epidemiological info here. 

Line 342. Reformulate this phrase, the “exactly” does not reflect here the definitive diagnosis concepts.

Line 343 – 344. Also this phrase does not make much sense. Reformulate. 

347. Complied with what?

Line 354. Expand here. The way this information is delivered is not clear to understand the bottom line and objective of the authors. 

Figure 6. Harmless does not mean much in infections diseases field. Please, specify what harmless means here. 

Author Response

We are very grateful for time reviewers invested in reviewing this paper (American foulbrood - old and always new challenge- vetsci-1999546) and constructive criticisms and corrections you recommended in order to improve our manuscript.

We took into account all suggestions, and we hope you will find that quality of manuscript is improved. In general, according to reviewers suggestions, particular mistake are corrected. Thus, we believe the manuscript is rearranged to meet requirements of reviewers. Specific corrections and comments are presented in the table below, and all changes are highlighted in the manuscript.

The article is very interesting and gathers important information on current knowledge of such important disease. However, some work needs to be done on the language and citation of appropriate scientific papers. Moreover, the article could benefit from more critical assessment of where the research in the field is going in terms of prevention of infection.  More detailed review is as follows:

  • Thank you for this comment

Maybe the abstract needs to refer more to the entire take home message of the paper if more is added on critical assessment of novel steps to prevent the disease.

Line 34 – add a reference for the OIE.

  • Yes, its added.

Line 41. Space missing after the references

  • - Yes, its added.

Line 44-48. Consider making this 2 phrases instead of one. It is long and disconnected.

  • This part is removed.

Line 62 – 63: use parenthesis instead of [] and throughout the text.

We used [] according to Introduction for authors.

Is there any specific selective media for P. larvae? I am assuming that the blood agar is not specific for the growth of P. larvae.

Line 69-70: Results of what?

Line 91-92: what is the gold standard technique for diagnosis of foulbrood? I think the paper would be more complete if some attention is given to the limitations of diagnosis and improvements needed. For example, should the gold standard be culture-dependent combined with further molecular techniques to determine genotypes? Is there any specific selective media for P. larvae? I am assuming that the blood agar is not specific for the growth of P. larvae.

  • Yes, Columbia blood agar is not the medium of choice for isolation, but it can be used, and here we have demonstrated the growth and appearance of colonies. Lines 389-391 list the selective media for the isolation of P. larvae. ‚‚In addition, there are several selective media for P. larvae culture: Paenibacillus larvae agar (PLA), MYPGP agar, BHIT agar (brain–heart infusion medium supplemented with thiamine), J-agar and CSA (Columbia sheep-blood agar )‚‚

Line 102-105. Please, rephrase. This statement is long, confusing, and not scientific. –

Yes- its corected.

Line 107-110. Same here, please rephrase and cite references to back up these statements.

Yes- its corected.

Line 118. Missing a space

  • Yes- its corected.

Line 119. The bees are not causative agents, they might be called vectors.

  • Yes, its corrected.

Line 128. Replace “get the disease” with scientific language.

  • Yes, its added.

Line 136. Please revise this sentence. They injected honey bee?

  • This is removed from text.

Line 140. Space missing

  • Yes, its corected.

Line 142-164. Please, revise language. Here again, language is not scientific and references are missing to support content.

  • In order to ensure the English language is of sufficient quality, manuscript has been edited by Dr Sheryl Avery of Avery Buncic Scientific & English Editorial Services (ABSeeS).

Line 189-197: What are the early signs they are looking for? Are there any surveillance sample collection and testing being done? What is the recommendation?

- Corrected

Line 221. Maybe expand on the special equipment. It is not clear the importance of this phrase here.

- Corrected

Line 223. Double “and” and space missing

  • This is removed.

Line 237. space missing

  • This is corrected.

Do you have references for your sampling instructions ?

- Entered, OIE manual

Line 269-273. The development of antimicrobial resistance and the risk of antimicrobials in the honey is a VERY important part of the story. Maybe expanding on it would be ideal. This phrase is very long and should be expanded to reflect the importance of the issue.

- Expanded and Explained

Figure 5 is not mentioned in the text. More information on the PCR diagnostic tool is needed. For example, what is the gene that the PCR is amplifying? Also, I don’t understand the legend. Is the 3737 a strain?

  • Corrected

Line 285 and 289. Missing spaces

  • Yes, its added.

First 3 paragraphs of the differential diagnosis: here again, more references and careful language is needed.

  • Yes, its corrected.

Interchangeable use of honeybee and honey bee. Stick to one format.

  • Yes, it's corrected in the text.

AFB prevention: is the disease notifiable? Is data on the prevalence available? It would be a good epidemiological info here.

            - Corrected

Line 342. Reformulate this phrase, the “exactly” does not reflect here the definitive diagnosis concepts.

Line 343 – 344. Also this phrase does not make much sense. Reformulate.

  1. Complied with what?

            - Corrected,  deleting

Line 354. Expand here. The way this information is delivered is not clear to understand the bottom line and objective of the authors.

            - Expanded, clarified

Figure 6. Harmless does not mean much in infections diseases field. Please, specify what harmless means here.

            - Yes, it was corrected

***On lines 492-493 and 522-523 duplicate reference ‚‚Stephan, J.G.; de Miranda, J.R.; Forsgren, E. American foulbrood in a honeybee colony: spore-symptom relationship and feedbacks between disease and colony development. BMC ecology. 2020, 20(1), 1-14. https://doi.org/10.1186/s12898-020-00283-w‚‚ with lines 522-523 deleted (29 and 41)

*** According to your advice and suggestions, eight new references were added to the work (27, 37, 51, 53, 55, 56, 57, 58)

We would love to thank you for allowing us to resubmit a revised copy of the manuscript. We believe that the revised paper is more comprehensive, and we do hope that it can meet your expectation.

Sincerely,

Jelena Ćirić, DVM, PhD

Round 2

Reviewer 1 Report

1. Introduction

Line 40-41

It is stated that only few data on the prevalence of AFB are available to date. The MLST data known since 2014 (Morrissey et al., 2014) were not considered here. However, samples collected worldwide were tested for P. larvae by Morrissey and colleagues.

This section has not changed. The reference Morrissey et al., 2014 is present in the reference list. However, it is not listed here.

2. American foulbrood and the etiology of the causative agent of the disease

Line 51

It is described that P. larvae was previously classified as Bacillus larvae. That’s correct. The intermediate step of the classification to Paenibacillus larvae subsp. larvae and Paenibacillus larvae subsp. pulvifaciens was still not described. This would also not be absolutely necessary if this designation had not been listed in Table 1.

Line 69-70

Djukic et al. (2014) was listed as a reference for the fact that different genotypes are known for P. larvae. In this paper, the sequencing of the entire genome of two P. larvae strains and the subsequent annotation of the genomes is described and the results are explained and discussed. Genotyping of P. larvae is not an issue here. The references given have not been changed at this point.

Line 70-73

The two successive sentences have the same meaning. No sentence was deleted, although the reply letter says otherwise.

Line 76-77

The authors explain that some P. larvae genotypes cause the same symptoms. Only some? What are the symptoms in strains of the other genotypes? Just because the beekeeper or veterinarian sees some symptoms rarely does not mean that they do not occur.

The sentence was not changed, although the reply letter says otherwise.

Table 1

Image 1 has become a table. Despite this, the classification of P. larvae ERIC III and IV in P. larvae subsp. pulvifaciens is still included. This classification is no longer current. The ERIC III and IV genotypes are also P. larvae (Genersch et al., 2006).

In addition, the data on LT100 of P. larvae ERIC II is still missing. This has already been determined. Adding my comment from my review there is absolutely not correct. It was a request to find out this information and to add it to the table.

Line 91-92

Genersch (2010) was given as a reference for a false negative diagnosis due to aqueous content. This reference does not describe this statement. The reference has not been changed here, although the reply letter says otherwise.

Line 104-105, 110-111

Authors need to express themselves more clearly, so that their presumably correct statements are also correct.

An infection does not develop from the larval food or from the uncapped brood, as described here. The larval food, for example, would have to be already contaminated.

The sentence was not changed, although the reply letter says otherwise.

Line 112-113

A reference to this statement is completely missing. The American Foulbrood is a brood disease. Adult bees are not susceptible to this disease.

This comment has not been taken into account in the text. There was no adjustment.

Line 134-136

The requested reference was not included.

Line 137

A wrong word has been used here, which makes the statement of the sentence inaccurate. P. larvae do not produce more virulence factors. If that were the case, a comparison parameter would have to be included. P. larvae produce many virulence factors.

The statement has not been changed. So it's still wrong.

Line 146-147

Likewise, here too a statement is presented completely incorrectly, due to the wrong choice of words. Bees are not pathogens. They only can transmit the spores of the pathogen.

The statement has not been changed. So it's still wrong.

3. Clinical signs of AFB

Line 168-169

Why should changes of the larva (colour, consistency, etc. ) not be preserved? These are not actually short-term effects in a P. larvae infection.

The statement was not changed or explained in more detail.

Line 172-177

The question remains open as to what is used to determine such an exact time. The given reference does not describe this issue.

Line 211-2013

The wrong reference is still included.

Figure 3B

A reference to figure 3B is not included in the text.

4. AFB control

Line 257-260

The answer to my comment was “Yes!”. What is that supposed to mean? There was no question and nothing was changed accordingly in the text.

Line 326-329

There is no change in the text.

Figure 5

I stick to it, no own results belong in a review. A PCR detection of P. larvae has been described many times. Appropriate references can be given.

Also, this is not a good example. A mismatched marker was used. If I don't have the material needed for an experiment, I can't do it like this.

5. AFB differential diagnostics

Line 337-338

I can still imagine confusion of AFB with EFB and sacbrood. How this can happen with an infestation of varroa is not clear to me. Or is a DWV infection meant here?

6. AFB prevention

Line 409-411

The existence of an association of beekeepers does not automatically produce healthy colonies.

Line 423

The larvae spores described here do not exist. Probably P. larvae spores are meant.

There was no corresponding adjustment in the text.

Author Response

Dear,

We are very grateful for time reviewers invested in reviewing this paper (American foulbrood - old and always new challenge- vetsci-1999546) and constructive criticisms and corrections you recommended in order to improve our manuscript.

We took into account all suggestions, and we hope you will find that quality of manuscript is improved. In general, according to reviewer’s suggestions, particular mistake is corrected. Thus, we believe the manuscript is rearranged to meet requirements of reviewers. Specific corrections and comments are presented in the table below, and all changes are highlighted in the manuscript.

Introduction

Line 40-41

It is stated that only few data on the prevalence of AFB are available to date. The MLST data known since 2014 (Morrissey et al., 2014) were not considered here. However, samples collected worldwide were tested for P. larvae by Morrissey and colleagues.

This section has not changed. The reference Morrissey et al., 2014 is present in the reference list. However, it is not listed here.

  • Corrected, Morrissey et al., 2014 is added.

  1. American foulbrood and the etiology of the causative agent of the disease

Line 51

It is described that P. larvae was previously classified as Bacillus larvae. That’s correct. The intermediate step of the classification to Paenibacillus larvae subsp. larvae and Paenibacillus larvae subsp. pulvifaciens was still not described. This would also not be absolutely necessary if this designation had not been listed in Table 1.

  • Corrected, deleted larvae subspecies pulvefaciens in table 1 (P. larvae)

Line 69-70

Djukic et al. (2014) was listed as a reference for the fact that different genotypes are known for P. larvae. In this paper, the sequencing of the entire genome of two P. larvae strains and the subsequent annotation of the genomes is described and the results are explained and discussed. Genotyping of P. larvae is not an issue here. The references given have not been changed at this point.

  • Corrected, deleted reference 22 from the list of references (Djukić et al. (2014))

Line 70-73

The two successive sentences have the same meaning. No sentence was deleted, although the reply letter says otherwise.

  • Corrected, first sentence deleted

Line 76-77

The authors explain that some P. larvae genotypes cause the same symptoms. Only some? What are the symptoms in strains of the other genotypes? Just because the beekeeper or veterinarian sees some symptoms rarely does not mean that they do not occur.

The sentence was not changed, although the reply letter says otherwise.

  • Corrected, deleted some.

Table 1

Image 1 has become a table. Despite this, the classification of P. larvae ERIC III and IV in P. larvae subsp. pulvifaciens is still included. This classification is no longer current. The ERIC III and IV genotypes are also P. larvae (Genersch et al., 2006).

In addition, the data on LT100 of P. larvae ERIC II is still missing. This has already been determined. Adding my comment from my review there is absolutely not correct. It was a request to find out this information and to add it to the table.

  • Corrected, deleted and in table 1, added data for LT100, strain ERIC II

Line 91-92

Genersch (2010) was given as a reference for a false negative diagnosis due to aqueous content. This reference does not describe this statement. The reference has not been changed here, although the reply letter says otherwise.

  • Corrected, deleted

Line 104-105, 110-111

Authors need to express themselves more clearly, so that their presumably correct statements are also correct.

An infection does not develop from the larval food or from the uncapped brood, as described here. The larval food, for example, would have to be already contaminated.

The sentence was not changed, although the reply letter says otherwise.

  • Corrected, replaced sentence

Line 112-113

A reference to this statement is completely missing. The American Foulbrood is a brood disease. Adult bees are not susceptible to this disease.

This comment has not been taken into account in the text. There was no adjustment.

  • Corrected, replaced sentence

Line 134-136

The requested reference was not included.

  • Corrected, added references

Line 137

A wrong word has been used here, which makes the statement of the sentence inaccurate. P. larvae do not produce more virulence factors. If that were the case, a comparison parameter would have to be included. P. larvae produce many virulence factors.

The statement has not been changed. So, it's still wrong.

  • Corrected

Line 146-147

Likewise, here too a statement is presented completely incorrectly, due to the wrong choice of words. Bees are not pathogens. They only can transmit the spores of the pathogen.

The statement has not been changed. So it's still wrong.

  • Corrected, replaced sentence

  1. Clinical signs of AFB

Line 168-169

Why should changes of the larva (colour, consistency, etc. ) not be preserved? These are not actually short-term effects in a P. larvae infection.

The statement was not changed or explained in more detail.

  • Corrected, replaced sentence

Line 172-177

The question remains open as to what is used to determine such an exact time. The given reference does not describe this issue.

  • We tried to clarify, but we have a problem to find this line.

Line 211-2013

The wrong reference is still included.

- We tried to clarify, but we have a problem to find this line

Figure 3B

A reference to figure 3B is not included in the text.

  • Corrected, reference included

  1. AFB control

Line 257-260

The answer to my comment was “Yes!”. What is that supposed to mean? There was no question and nothing was changed accordingly in the text.

We tried to clarify, but we have a problem to find this line

Line 326-329

There is no change in the text.

We tried to clarify, but we have a problem to find this line

Figure 5

I stick to it, no own results belong in a review. A PCR detection of P. larvae has been described many times. Appropriate references can be given.

Also, this is not a good example. A mismatched marker was used. If I don't have the material needed for an experiment, I can't do it like this.

  • Corrected, deleted everything
  1. AFB differential diagnostics

Line 337-338

I can still imagine confusion of AFB with EFB and sacbrood. How this can happen with an infestation of varroa is not clear to me. Or is a DWV infection meant here?

  • Yes, we tried to clarify

  1. AFB prevention

Line 409-411

The existence of an association of beekeepers does not automatically produce healthy colonies.

  • Corrected, replaced sentence

Line 423

The larvae spores described here do not exist. Probably P. larvae spores are meant.

There was no corresponding adjustment in the text.

  • Corrected

***Two references, 22, Đukić et al., 2014 and 42 Reig et al., 2010, were deleted in the work and the list of references.

We would love to thank you for allowing us to resubmit a revised copy of the manuscript. We believe that the revised paper is more comprehensive, and we do hope that it can meet your expectation.

Sincerely,

Jelena Ćirić, DVM, PhD

Round 3

Reviewer 1 Report

First of all, I have received a document which obviously contained comments on the side of the page. These were invisible. I hope it did not contain any decisive information.

1.     Introduction

Line 74-76

As it is now described there, the spores are in the first instar. But it is the larvae that are at this stage. This needs to be adjusted accordingly.

2.     American foulbrood and the etiology of the causative agent of the disease

Line 139-140

White is given as a reference for Bacillus larvae. However, the year is missing and references 11 and 19 are referred to overall. This is not the correct reference.

Figure 1

It is described that in Figure 1A vegetative bacteria and spores are recognizable. There are already significant differences in size between bacteria and spores. What are bacteria in this picture and what are spores?

Line 157-160

The two successive sentences have the same meaning. No sentence was deleted, although the reply letter says otherwise.

Table 1

The table has been adapted according to the specifications. Now, however, the line “Species” is redundant and does not need to be listed.

Line 241-242

A corresponding reference is missing.

Line 246-248

Authors need to express themselves more clearly, so that their presumably correct statements are also correct.

An infection does not develop from the larval food or from the beehive, as described here. The larval food, for example, would have to be already contaminated.

The sentence was not changed, although the reply letter says otherwise.

Line 254-256

A corresponding reference is missing.

Line 265-267, 380-383

The meaning of the sentences is not clear. As it is now described, the larva is cleaned by the nurse bees. This is not the case. The nurse bees remove the dead larva and then clean the brood cell.

3.     Clinical signs of AFB

Line 384-385

Why should changes of the larva (colour, consistency, etc. ) not be preserved? These are not actually short-term effects in a P. larvae infection.

The statement was not changed or explained in more detail.

Line 388-393

The question remains open as to what is used to determine such an exact time (20 days). The given reference Ebeling et al. (2016) does not describe this issue.

Line 459-460

The wrong reference was listed here.

Line 466-467

Larvae have been described to die between day 3 and 12. Then it is not possible for all the larvae to die before cell capping. Opposition within a sentence.

Line 467-470

That's too imprecise. It is not true that all larvae infected with P. larvae ERIC I die only after capping the brood cell.

Line 474-477

AFB is neither a bacterial strain nor a genotype. AFB is the disease and not the pathogen. Technical terms need to be handled much more precisely.

Line 525-526

repetition of the sentence

Line 527-529

It is not true that ERIC I genotype is more common than the other genotypes (see MLST data). It is only easier to detect the typical AFB symptoms in an ERIC I infection than in an ERIC II infection.

Line 529-530

The statement of line 470-471 was repeated.

Line 530-532

The wrong reference is still included.

Figure 3B

A reference to figure 3B is not included in the text.

4.     AFB control

Line 561-643

The methods specified by the responsible veterinarian must be used. In such cases, it is not possible to make one's own decisions everywhere. There are also cases in which it is possible to carry out an artificial swarm process.

Line 635-637

In order to detect a P. larvae infection, both the positive laboratory evidence and typical AFB symptoms in the colony must be demonstrated. If one point cannot be achieved, the colony is declared uninfected. But that doesn’t mean that an infection has ended, it means that there is no infection.

Line 664-740

This section is far too general. Not every laboratory examines every sample material. Sampling and dispatching samples must be carried out according to the specifications of the laboratory.

Line 745

There is no vegetative form of AFB. There is no clear distinction between disease and pathogen.

Line 758-760

Honey contaminated with P. larvae spores is still marketable and can be sold. This statement cannot be supported by the reference given.

5.     AFB differential diagnostics

Line 762-763, 770-772, 874-876

I can still imagine confusion of AFB with EFB and sacbrood. How this can happen with an infestation of varroa is not clear to me. Or is a DWV infection meant here?

Varroa parasitizes pupae. Larvae cannot die from it. If such a statement is made, it must also be explained well. The reference provided does not support the statement. In general, original scientific references should be used.

Line 868-870

In AFB, larvae also die in uncapped cells. This has been described several times in this manuscript.

Line 871-873

This is not described in the given reference. This reference does not cover European foulbrood.

Line 878-880

unnecessary repetition

Author Response

Respected Editor and Reviewers,

We are very grateful for time reviewers invested in reviewing this paper (American foulbrood - old and always new challenge- vetsci-1999546) and constructive criticisms and corrections you recommended in order to improve our manuscript.

We took into account all suggestions, and we hope you will find that quality of manuscript is improved. In general, according to reviewers suggestions, particular mistake are corrected. Thus, we believe the manuscript is rearranged to meet requirements of reviewers. The all changes are highlighted in the manuscript.